# Government, Premier Leader and Small Lakes' People Vis-a-Vis Lake Governance

Bing Baltazar C. Brillo 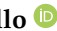

Institute for Governance and Rural Development, College of Public Affairs and Development, University of the Philippines Los Baños, Los Baños 4031, Laguna, Philippines; bcbrillo@up.edu.ph

**Abstract:** Small lakes within social sciences' conceptualisation are mostly wanting, less anchored and seldom scrutinised in academic literature as opposed to large lakes and natural sciences. Essentially, small lakes, from a social sciences' perspective, are about people connecting and enhancing through lake governance. Thus, the main argument is that a small lake's people must accept, broaden and elevate the prospect of lake governance by focusing on and embracing the central concepts of government—the most compulsory and crucial constituent—as well as premier leader—the most pre-eminent and imperative function. Accordingly, lake governance refers to engaging with and intervening in the collective people of a small lake, to undertake economic development, pursue ecological conservation and manage government. Government refers to steering a small lake's people towards emphasising executive authority and decision-making power, whether through solutions, policies, regulations and/or implementations. The premier leader refers to the person presiding over a small lake's people in the critical aspects of resoluteness—in establishing and sustaining the rules—and decisiveness—in settling and determining a community's issues. Overall, as small lakes' people are political, lake governance is consequential, and a government/premier leader is evidently the most efficacious outcome, whether for addressing problems, choosing decisions or ameliorating society.

**Keywords:** government; governance; lake; lake governance; leader; small lake

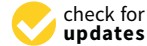



## 1. Lakes' People, Small Lakes and Social Sciences

Lakes exist and are significant. Lakes are innumerable on Earth, from tropical to temperate to polar regions, and they are valuable for humanity, whether they be natural lakes, artificial lakes (i.e., man-made reservoirs), freshwater lakes or saltwater lakes (i.e., saline lakes). In abiotic and biotic terms, lakes are essential to natural processes, such as nutrient cycles and water cycles (i.e., the hydrologic cycle), and for sustaining biodiversity, such as habitat rehabilitation and fish sanctuaries. For people, lakes are necessary for basic uses, such as water sources, food supplies, recreation and transportation, as well as considerable utilities, such as water supplies, agriculture irrigation, hydropower, aquaculture/farming and the tourism industry. As a water resource, lakes are integral to flora–fauna life and intrinsic to human survival. By definition, a lake simply means an interior waterbody contained in a basin and often entirely surrounded by land. A lake's concave surface area is substantial on the water's volume and depth, and, usually, it permits water to flow into inlets and/or outlets (i.e., going in and out of the lake). A lake's water sources are usually fed via rivers/streams, aquifers/springs or rainfall/snowfall, and discharged by natural outflow, evaporation and underground seepage.

Water use is the most important aspect but a predicament for a lake's people, whether due to exorbitant consumption, withdrawals, diversions or contractions, excessive degradation, pollution or contamination (e.g., World Lake Vision Committee 2003; International Lake Environment Committee Foundation 2005; World Wildlife Fund 2021b). In terms of global water volume, lakes constitute approximately 176,400 cubic kilometres (0.013 percent), where freshwater lakes are roughly 91,000 cubic kilometres (0.007 percent) and



saltwater (or saline) lakes are approximately 85,400 cubic kilometres (0.006 percent) (Shiklomanov 1993; US Geological Survey 2018a). In surface freshwater distribution, lakes are 20.9 percent and rivers are 0.49 percent; both are vital, since they are in use by lake people every day (Shiklomanov 1993; US Geological Survey 2018b; World Wildlife Fund 2021a). Although lakes comprise a diminutive percentage of the world, they contribute significantly in using water, enough for the people to sustain and flourish. Lakes' peoples' use of water is currently problematic. Over 2 billion people worldwide are facing constant water stress, where the corresponding demand of humans for freshwater is undersupplied. Approximately 4 billion people are experiencing persistent water scarcity, where the related supply volume of freshwater is insufficient (CEO Water Mandate 2014; Mekonnen and Hoekstra 2016; UNESCO World Water Assessment Programme 2019). In agriculture, water consumption in terms of yearly water withdrawals from lakes and rivers is extreme (approximately 70 percent globally and above 90 percent in some developing countries) in contrast to other sectors, which generates concern on the dispute in utilisation and sustainability (AQUASTAT 2015; UNESCO World Water Assessment Programme 2017; Biswas and Tortajada 2010).

Lakes' peoples' activities have repercussions on water resources and communities. Human pursuits and manipulations render benefits attributed mostly to necessity and/or progress in lakes (e.g., fish production, human settlements, agriculture industry and urban development). However, these activities also have ramifications for water resources. Unquestionably, the positive effects of lakes have long been recognised, but the adverse effects have also acknowledged that lakes manifest numerous problems. Human activities, whether incredibly exploitative and/or characterised by unrestrained practices, have severe consequences on lakes, such as undesirable fishing practices, excessive effluent discharges, immoderate amounts of nutrients, unfettered contamination, unconstrained water extraction and the introduction of exotic species (e.g., Kira 1997; Jorgensen et al. 2003; Aralar et al. 2005; US Geological Survey 2019; Jenny et al. 2020). Moreover, people's impacts reach beyond water sources, since a drainage system (i.e., watershed, drainage basin and catchment area) and the contiguous or associated area (i.e., littoral zone, riparian zone and upland zone) have broad complications for lakes. Overall, people–lake issues have been recognised and extensively discussed by the globally scaled study by the Global Environment Facility—the Lake Basin Management Initiative of 28 significant lakes from the initial to the recent World Lake Conferences (World Lake Conference 1984, 2018; International Lake Environment Committee Foundation 2005; Aralar 2013).

Quantities of lakes are considerable in the aggregate. Large lakes and small lakes exist in the millions throughout the world, and the significance of both is acknowledged. Juxtaposed with small lakes, there are not many large lakes, but they are normally substantial in water volume, area size and depth. Of the 253 accepted large lakes, approximately 90 percent of the water capacity (roughly 179,000 cubic kilometres) and surface area (approximately 1,400,000 square kilometres) are possessed throughout the world (Herdendorf 1990; LakeNet 2004b, 2004c; Cael and Seekell 2016; International Association for Great Lakes Research 2021). Typically, studies on large lakes have been entrenched and contributed substantially to the academic literature. For large lakes, the natural sciences, spearheaded by the disciplines of biology, chemistry and physics, have produced considerable scholarly works, particularly in the area of limnology, hydrology and ecology (e.g., thermal stratification, water chemistry/quality, surface hydrology, fisheries science and ecosystem services). For large lakes, the social sciences, led by the disciplines of sociology/anthropology, geography and economics, have also shared academic works, especially in the area of human, society and development studies (e.g., environmental sociology, human ecology, human geography, development studies and development economics). Generally, large lakes are overrepresented in studies, an occurrence attributable to the immensity of their functions and their considerable consequences for water and people (e.g., Downing et al. 2006; Oertli et al. 2009; Downing 2010; Brillo 2015c, 2016c). Contrastingly, small lakes are different when examining the academic literature—immensely sufficient in natural sciences and

exceedingly less common in the social sciences. For example, using the words "small lakes" in Google Scholar shows 79,700 article results with eight related searches (i.e., ponds, sediments, zooplankton, Southern Finland, trout, large–small lakes, water quality and phytoplankton), which discloses that studies on them are almost exclusive to the natural sciences (Google Scholar 2021e). With this, scholarly works on small lakes are plentiful globally, yet are mostly confined to the sphere of natural sciences studies and highly marginal in the realm of social sciences studies. In other words, overall, due to the lack of social-sciences-based and small-lake studies, the natural-sciences-based and large-lake studies provide a partial picture.

A good example is the case of the Philippines, where the large lakes (i.e., Laguna de Bay (93,000 hectares), Lanao Lake (34,000 hectares), Taal Lake (23,420 hectares) and Mainit Lake (17,340 hectares)) have abundantly supplied the output, while the small lakes have inconsiderable scholarly studies. For instance, using the words "Laguna de Bay" in Google Scholar, the country's largest lake, has 4880 articles results and 16 related group searches (i.e., fish, water quality assessment, lake, heavy metal, eutrophication, Nile tilapia, aquaculture, carp, oreochromis niloticus, water pollution, algal bloom, manila, knife fish, metals and fisheries) up to the present (Google Scholar 2021c); on the other hand, small-lake studies are paltry, with approximately three-quarters of them in the Philippines either unaccounted for, paid minimal attention, and/or not yet studied (e.g., Brillo 2015c; Brillo et al. 2019). The record of lakes in the Philippines (including major lakes and minor lakes) is the following: the LakeNet Global Lake Database has 42 lakes, the National Mapping and Resource Information Authority has 78 lakes, Wikipedia's list of lakes in the Philippines has 94 lakes, the DENR—Biodiversity Management Bureau has 105 lakes (only in Luzon, the Philippines) and Brillo *Asia-Pacific Social Science Review* has 198 lakes (LakeNet 2004a; National Mapping and Resource Information Authority 2014; DENR—Biodiversity Management Bureau 2014; Brillo 2015c; Wikipedia 2021b). Overall, the information shows that lake studies in the Philippines need to be expanded, particularly embracing small lakes. Studies on small lakes in the Philippines have incrementally escalated over the years, often published in conference proceedings (e.g., Aralar et al. 2005; Aralar 2013) and mostly focused on natural sciences works, specifically limnology and aquaculture studies (e.g., Cuadrado et al. 2019; Mendoza et al. 2019; Ballares et al. 2020; Guevarra et al. 2020). However, the Philippines' small-lake studies are typically minimal in terms of social sciences works (Brillo 2015c, 2017b, 2017c). Social-sciences-based small-lake studies have slowly increased since 2016 in the Philippines; nevertheless, they are limited, and thus more are still needed (e.g., Brillo 2016a, 2016d, 2020b; Brillo and Boncocan 2016, 2017a; Cudera et al. 2020).

From the past to the present, the precise definition of small lakes is still vague, as there is a cliche that there is no universally accepted denotation on small lakes (versus large lakes) from scientists to disciplines. A small lake has been described chiefly in natural sciences, such as by limnologists, hydrologists, fishery scientists and geoscientists, and sometimes mentioned as a minor lake or a pond. The extant definitions are several but remarkably diverse, from the properties to the dimensions of a body of water (e.g., Moss et al. 1996; Bramick 2002; Williams et al. 2004; Granger and Hedstrom 2011; Trondman et al. 2016; Williamson et al. 2018). For example, in size, the magnitude of the area is highly varied, usually from 1 to 15 hectares for ponds to 25–400 hectares for small lakes (e.g., Williams et al. 2004; Søndergaard et al. 2005; Downing et al. 2006; Cech 2018; Google Scholar 2021b; Wikipedia 2021a, 2021c). In contrast to the social sciences, there is less attention on clearing the denotative definition of small lakes. This deficiency has implications for social sciences as the "people's study", since small lakes have typically different dynamics and engagements than those of large lakes. For instance, small lakes are generally under few domains, resources and inhabitants, while large lakes are commonly subjected to more domains, resources and inhabitants. With this, it is paramount to settle and operationalise a definition of small lakes in social sciences studies. Under this, small lakes are demarcated as water bodies with a surface area of 200 hectares or less (while large

lakes are above this threshold). This size limit justification is connotative and subjective, clutching and relaxing social sciences (against denotative and objective via natural sciences) and following the survey variance in lakes and small lakes' peoples' experiences and research (e.g., Brillo 2015c, 2017b, 2017d). In other words, the 200 hectares or less sense does not embrace the natural sciences perspective, but from the social sciences viewpoint it makes it straightforward to understand and favourable to designate the administrative and collective peoples of small lakes.

Broadly, small lakes have compelling grounds for undertaking academic work on them. First, small lakes have a shorter timeline in deterioration compared to large lakes. Intrinsically, the smaller dimension of the water's volume lessens the cushioning capacity in neutralising contaminants, making it more susceptible to degradation. Good examples here are abandoned open-pit mining lakes, such as Capayang Lake in Mogpog as well as Marinduque and Lubo Lake in Kibungan, Benguet, which precipitate a decline due to high heavy metal concentrations, posing hazards to human health and the environment. Second, small lakes have a diminutive supply of information on conservation and amelioration. Stockpiling details and facts are necessary conditions for managing and enhancing the many small lakes, especially those that are unexplored, remote or diverse. A good example is Ambulalacao Lake, one of the highest small lakes in the Philippines, which is diminutive in the scholarly literature. Third, small lakes are not standalone but are affected by other constituent systems. The inland water connects to complex components (e.g., groundwater systems, river systems, large-lake systems, drainage systems or man-made systems); thus, understanding the resources and sources is consequential. The archetypal instance, here, are the Seven Crater Lakes in San Pablo City, Laguna (i.e., Sampaloc Lake, Bunot Lake, Palakpakin Lake, Mohicap Lake, Pandin Lake, Yambo Lake and Calibato Lake), which are separated but linked by affiliative to the Laguna de Bay watershed. Fourth, small lakes are abundant but more prone to disappearance or extinction (e.g., drying or infilling) than large lakes. The small interior waters should be documented for preservation and posterity. A good case in point is Manlalayes Lake (i.e., the twin lake of Gunao Lake) in Dolores, Quezon, which dried out over a few decades before recording. Fifth, small lakes are associated with people and communities both near and/or distant. Inland water assets are essential in utilisation and catalytic in developing the collectivity, especially the impoverished populace. A typical case is Pandin Lake, in which the community with a local environmentalist group was able to institute and develop into an ecotourism enterprise (see Brillo 2015a, 2015b, 2016b, 2016e, 2017a).

The scholarly literature in the natural sciences has already argued since the 2000s that small lakes are indispensable in sustaining localised biodiversity and global processes, such as being cumulatively more biologically active and having more species taxa per unit area as well as a more intense carbon cycle pound-for-pound than large lakes (e.g., Lehner and Doll 2004; Downing et al. 2006; Hanson et al. 2007; Oertli et al. 2009; Downing 2010; Google Scholar 2021e). Social sciences must correspond, complement and engage with the natural sciences, especially in regard to small lakes' issues of the utmost importance. Social sciences studies ought to profoundly connect to small lakes from the present unto the future. In the broadest sense, social sciences mean studying human beings and societies, whether that be their interactions, comportments and/or relationships. In small lakes, social sciences are about the peoples of the lakes, particularly concerning the collective group of the community, regardless of inhabitants, settlers and/or townspeople. In juxtaposition, the natural sciences deal fundamentally with life and the physical world, while social sciences cope primarily with people. Thus, although different, the two disciplines are securely bridged—interconnected in the understanding of and augmenting the progress of lakes. However, at their heart, the social sciences of lakes must supplement and advance the improvement of their peoples.

In the social sciences, people are crucial in small lakes on account of development and conservation. Development is broadly equated to economic development in addition to the enhancement, inclusiveness and sustainability of societal endeavours or activities,

whether those be agricultural, production or commercial assets of lakes. In development, a populace uses a lake's resources owing to their economic significance to improve, expand and maintain living and community conditions, from livelihoods and labourers to fisheries–aquaculture, agriculture, tourism and manufacturing. Conservation is broadly considered to be ecological conservation, the maintenance, protection and restoration of societal endeavours or activities, whether of the aquatic or terrestrial environment, as well as of the biodiversity of lakes. In conservation, a populace, whether nearby or distant, applies and impacts the ecological significance of lakes directly and indirectly on their biotic and abiotic constituents. As an aphorism, a lake and its people's quandary are indisputably integrated and cannot secede between development and conservation—they are not against each other but in concert. If one is asunder from the other, then the task would be inadequate in grasping the dilemma of the lake and the community. One standalone point would not capture the whole complexity, as the population and the small lake have a multifarious impediment in utilising the undertaking of development and executing the pursuit of conservation.

Development and conservation in lakes are present, but their demands are often uneven (or worse, inharmonious); one demand corresponds more with gains, while the other demand corresponds less. The usual instance is the progression of a community in a lake, where the people's development demands are the remedy and the conservation demands are left behind. In a small lake, a classic example is the case of Sampaloc Lake: as the people embraced economic development from the 1980s to 1990s, the communities were exorbitantly swelled in tilapia farms, housing, informal settlements, restaurants and bars. Consequently, ecological conservation was neglected as the small lake's problems manifested, such as water pollution, fish kills, algal blooms, water hyacinth proliferation and slow tilapia growth (see Brillo 2016e, 2017d). Furthermore, development and conservation in small lakes are typically exacerbated in their demands due to the fact of being "minuscule" values. Economically and ecologically, the minimum values are ordinarily translated into less attention, less determination and/or less priority in a small lake. In other words, small lakes, overall, have few populations, a limited voice, restricted finances, narrow resources, marginal technologies and fewer international organisations, which usually transcribe to irresolute administration. A good example is the case of the Laguna Lake Development Authority's (LLDA) Management and Development Plan (MDP) between the Seven Crater Lakes vis-a-vis Laguna de Bay. Since 1983, with the LLDA's jurisdiction of the seven small lakes, up to the 2000s the MDP had not been implemented, primarily since Laguna de Bay was the focal point, being the biggest lake and the principal concern (while the small lakes were not). Therefore, the LLDA took approximately 32 years to launch the MDP of the seven lakes (specifically Pandin Lake, Sampaloc Lake and Yambo Lake) starting from 2015 onwards (see Brillo 2017a; Brillo et al. 2019). With these, finding the equilibrium of development and conservation was challenging, especially in making ameliorative, inclusive and sustained conditions of small lakes and their communities. Thus, these representative cases are the reason why the main constituents of lakes—the people, development and conservation—are crucial in endowing governance.

Overall, this study discusses and urges the addressment of a critique of small lakes and lake governance. The article aspired to look into a central concept for small lakes' people in the social sciences, particularly lake governance, its most crucial constituent—government—and its most decisive function—premier leader. The main argument is that social sciences in lakes, especially small lakes, must accept, broaden and elevate lake governance by focusing on and embracing the concepts of government and premier leader. This study encompasses the literature gaps to connect, enhance and expand the ideas of social sciences in lake governance vis-a-vis government, premier leader and the people to obtain a bigger picture of small lakes. The conceptualisation among them is mostly wanting, less attached and seldom scrutinised in the academic literature. The compositional concepts are confronted and synthesised to amplify the knowledge base in the social sciences and situate lake governance in lake scholarship. This is part of the deep-rooted desire to explore

and perpetuate the academic rationale of small lakes. Therefore, this paper delineated the discourse as follows: Section 1—Lakes' People, Small Lakes and Social Sciences; Section 2—Lake Governance to the Government and Small Lakes' People; Section 3—Government and Premier Leader in Small Lakes' People; and Section 4—Concluding Remarks.

## 2. Lake Governance to the Government and Small Lakes' People

Governance is cliché—since this contemporary name was conceived and introduced in the social sciences in the 1990s, the concept has been disseminated and embraced globally by academics, practitioners, businesses and politicians. Governance's label has been enthusiastically and customarily used as an umbrella term among academic disciplines, governmental agencies, administrative companies and international organisations. At present, the concept of governance is omnipresent, applying to a wide range of categories, such as public governance, private governance, nonprofit governance, global governance, corporate governance, environmental governance, land governance, internet governance, regulatory governance, participatory governance and collaborative governance. This implies that this concept is too large, such that it captures many aspects with different applications. Loosely, governance is a modus operandi—a way, action, manner, means or practice of governing people, whether that be in an organisation, a municipality/city or a state. This terminology has been deliberated, delineated and explicated from the past to the present.

Routinely, the following are the recognised definitions of governance: (1) Governance is "the manner in which power is exercised in the management of a country's economic and social resources for development" (The World Bank 1994); (2) "Governance is the sum of [the] many ways individuals and institutions, public and private, manage their common affairs. It is a continuing process through which conflicting or diverse interests may be accommodated and co-operative action taken. It includes formal institutions and regimes empowered to enforce compliance, as well as informal arrangements that people and institutions either have agreed to or perceive to be in their interest" (The Commission on Global Governance 1995); (3) governance is "the formulation and execution of collective action at the local level. Thus, it encompasses the direct and indirect roles of formal institutions of local government and government hierarchies, as well as the roles of informal norms, networks, community organisations, and neighborhood associations in pursuing collective action" (Boadway and Shah 2009); (4) governance is "the traditions and institutions by which authority in a country is exercised", which include "the process by which governments are selected, monitored and replaced; the capacity of the government to effectively formulate and implement sound policies; and the respect of citizens and the state for the institutions that govern economic and social interactions among them" (Kaufmann et al. 2010; Worldwide Governance Indicators 2021); (5) "Governance refers, therefore, to all processes of governing, whether undertaken by a government, market, or network, whether over a family, tribe, formal or informal organisation, or territory, and whether through laws, norms, power or language" (Bevir 2012); (6) "Governance refers to the exercise of political and administrative authority at all levels to manage a country's affairs" (UN System Task Team on the Post-2015 UN Development Agenda 2012); (7) "Governance as a government's ability to make and enforce rules, and to deliver services, regardless of whether that government is democratic or not" (Fukuyama 2013); and (8) "Governance is the systems and processes that ensure the overall effectiveness of an entity—whether a business, government or multilateral institution" (UN Global Compact 2021).

The Governance's concept is not many in lakes, and those where it is present are anchored mainly in water governance (e.g., Biswas and Tortajada 2010; Groenfeldt and Schmidt 2013; Brillo 2022). From the past to the present, the following are the authoritative definitions of water governance: (1) Water governance is "the political, social, economic and administrative systems in place that influence water's use and management", which means "essentially, who gets what water, when and how, and who has the right to water and related services, and their benefits" (Allan 2001; UNDP-SIWI Water Governance Facility

2021); (2) water governance is "the range of political, social, economic and administrative systems that are in place to develop and manage water resources, and the delivery of water services, at different levels of society" (Rogers and Hall 2003); (3) water governance is a polycentric governance system where "political authority is dispersed to separately constituted bodies with overlapping jurisdictions that do not stand in hierarchical relationship to each other" (Skelcher 2005; Huitema et al. 2009); (4) water governance is "the interaction of laws and other norms, institutions, and processes through which a society exercises powers and responsibilities to make and implement decisions [affecting lakes and their basin resources as well as their users] and to hold decision makers and implementers accountability" (Moore 2010); (5) water governance is the "range of political, institutional and administrative rules, practices and processes (formal and informal) through which decisions are taken and implemented, stakeholders can articulate their interests and have their concerns considered, and decision makers are held accountable for water management" (Organisation for Economic Co-Operation and Development 2018); and (6) water governance is "a combination of functions, performed with certain attributes, to achieve one or more desired outcomes, all shaped by the values and aspirations of individuals and organisations" (Jiménez et al. 2020).

The definitions of governance and water governance are plentiful, but their understanding, utilisation and application in lakes, particularly small lakes, are scant. Governance is a contested concept and usually endows too much conceptualisation. Water governance is designed for water matters and mainly deliberates for the universal—fitting most cases. Using both of them, therefore, would be restrictive of the focus on lakes, especially small lakes. Governance and water governance are too big to capture the specific distinctiveness and circumstances of small lakes, particularly in deploying different perceptions and contexts of lakes' people. In other words, it demands more refinement in looking at small lakes' people against governance and water governance as well as other distinct lakes (e.g., large vs. small lakes, densely vs. sporadically populated lakes, autonomous vs. transboundary lakes and isolated/remote vs. accessible/close-area lakes). The understanding of small lakes' people should be examined discretely, as lakes vary (from scenarios and situations to outcomes and implications), which do not suit all cases, until small-lake studies are sufficiently produced with which to standardise and apply liberally on a large scale. This is why it is better in small lakes (together with large lakes) to enhance the immense term of governance (and water governance) by metamorphosing and concentrating on the distinguished principle—lake governance.

The conceptualisation of lake governance is deficient and esoteric at present. This term is prevalently cursory and without an explicit explanation of its meaning in its application. In the literature on lakes, the concept of lake governance has seldom been scrutinised and mainly applies to two dominant studies—the water assessments of transboundary lakes (see Grover and Krantzberg 2018) and the documenting of small lakes in the Philippines (see Brillo 2015a, 2015b, 2015c, 2016a, 2016b, 2016c, 2016d, 2016e, 2017a, 2017b, 2017c, 2017d, 2020a; Google Scholar 2021d; Microsoft Bing 2021). Both of them minimally discuss the central concept in the literature on lakes, giving it insufficient attention in the discourse. To earn a better cognisance and whole perspective, the idea of lake governance should be elevated in the social sciences of lakes, particularly in small lakes, the people and the government. Accordingly, the concept of lake governance is defined as the political processes in which authority and power are exercised in the administration and management of lakes and their people for economic development and ecological conservation (Brillo 2022). Under this denotation, lake governance is about a lake's people, with the primary function of the government as well as development and conservation in operating effectively (see Figure 1). In other words, the collective people of a small lake are engaged in undertaking economic development, pursuing ecological conservation and managing/intervening in the government. The Government is the premier in lake governance since the nucleus of lake's people is the pre-eminence of decision making and executive, whether solutions, policies, regulations and/or implementations.

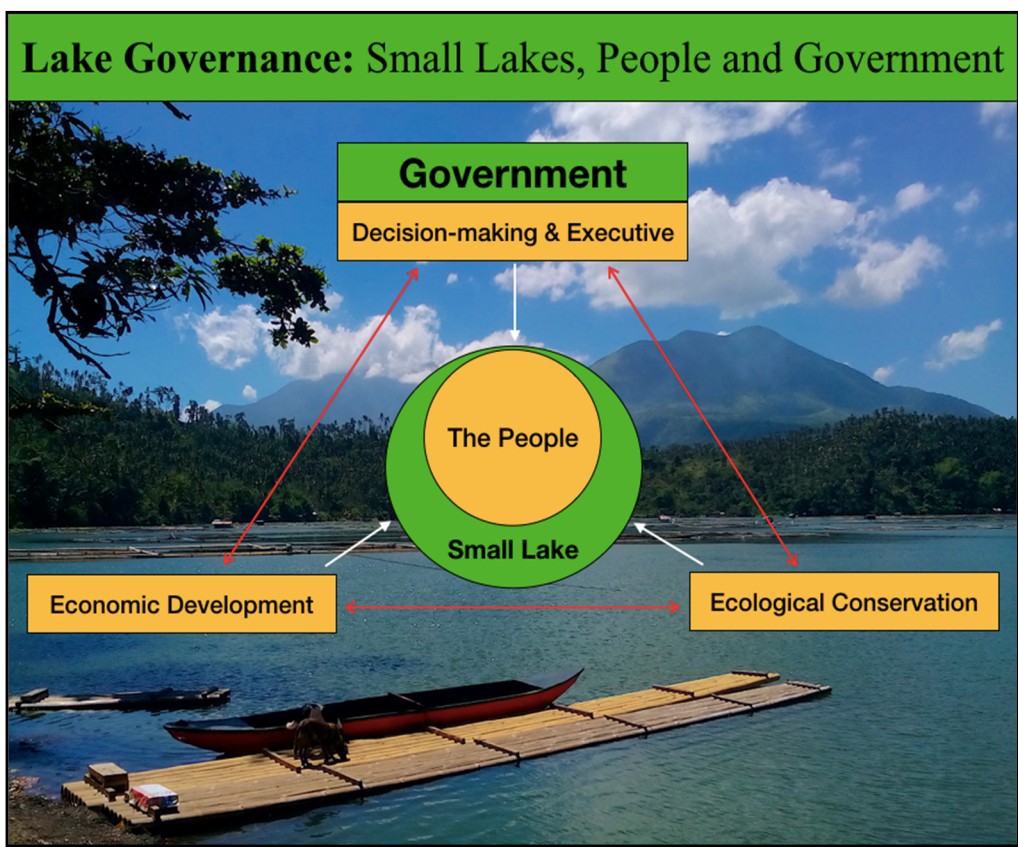

**Figure 1.** Lake governance: small lakes, people and government.

Broadly, the government is a formal institution with jurisdiction over a lake, being either the national government, governmental agencies, local governments (those of provinces, cities or municipalities) or barangays. In small lakes, the government commonly corresponds to the local government and is augmented by governmental agencies and barangays (e.g., Brillo et al. 2017a; Anastacio and Brillo 2020; Brillo 2020a). One anchored connection between the local government and the small lakes is with respect to financial resources. In small lakes, the local government usually provides the barangay and solicits governmental agencies to distribute funds; the barangay often has limited finances, and governmental agencies typically have capacitive finances. There are several examples of this subject matter in small lakes; for instance, in the case of Dagatan Lake, a small freshwater lake located in the San Jose barangay in the municipality of San Antonio, Quezon (Philippines). In Dagatan Lake, governmental agencies—the Department of Agriculture (DA) and the Bureau of Fisheries and Aquaculture Resources (BFAR)—have steadily supported and subsidised Dagatan Lake's restoration project, and the barangay—the Dagatan Lake Fisherfolk Association (DLFA)—has continuously been funded by the local government to protect and maintain the activities (see Brillo 2017b; Brillo et al. 2017b). Overall, this arrangement between the DA–BFAR and DLFA has constantly existed in this small lake.

In lakes, government and nongovernmental institutions, such as community organisations or nongovernmental organisations (NGOs), are essential, as they supplement each other. This is pivotal in small lakes, since the local government cannot meet all of the people's needs; thus, nongovernmental institutions are necessary. At worst, when the local government is absent or not operating consistently for a small lake's people, nongovernmental institutions then become more crucial. Nongovernmental institutions tend to supplant the tasks of the government in lakes. A classic example is the case of Pandin Lake, a small lake situated in the Santo Angel barangay, San Pablo City, Laguna (Philippines). The local government—the City Government of San Pablo—and the governmental agency—the Laguna Lake Development Authority (LLDA)—are the collaborating authorities of Pandin

Lake. Since the government was not active in developing the small lake before the 2000s, a nongovernmental institution—an NGO—became involved and helped the lake's people organise and launch a tourism enterprise—the Pandin Lake Tour project (Brillo 2016b; Brillo and Boncocan 2016). Although the tourism enterprise was eventually established, the government would inevitably embark and come into play in the small lake. Sustainability is a concern, considering that Pandin Lake's enterprise is afflicted by finance, accessibility, infrastructure and development management plan issues. With these obstacles, the city government and the LLDA gradually entered, engaged and intervened with the small lake's people in the 2010s (Brillo 2016b; Laguna Lake Development Authority 2014; Brillo 2020a). Notwithstanding the NGO's departure, the government's presence has slowly assisted and ameliorated Pandin Lake's situation. Thus, in the long run— government necessitates small lakes.

Establishing a government's presence in a lake is frequently costly. Usually, instituting a government in a lake requires overheads, such as installing effective organisations, enforcing policies, involving diverging people, utilising interventions, gathering information and sustaining funds (see Nakamura and Rast 2014; International Lake Environment Committee Foundation 2021). These running costs are considered when a small lake's people are impoverished and/or the municipality has a low income (e.g., a Fifth or Sixth Class Municipality). Despite it being demanding to establish a presence, the government has no choice but to move into the lake. The government is permanent compared to nongovernmental institutions, which are usually temporary for a long term at a lake (e.g., an NGO or a business can cover an issue, such as funds, once or twice, but not forever, as opposed to the government). Although it is difficult for the government to embark, it will slowly assert and direct the affairs of a lake's people when instituted. Thus, the government entails being persistent to become steady for a long time.

The government is the essence of the people in small lakes. The dynamics of small lakes have some variation in the inherence of and demands on the government compared to large lakes. Many small lakes are generally managed by the local government, supervised in one (sometimes two or three) municipality and operate small-scale development activities (e.g., limited aquaculture or tourism). In contrast, many large lakes are normally administered by a national governmental agency, preside in transboundary (sometimes two or more) provinces or countries and utilise extensive development projects (e.g., commercial fishing, agricultural irrigation or hydroelectric energy) (see Brillo 2017a, 2017d). With these, local government is vital and indelible in small lakes' challenges, especially in utilising resources or addressing peoples' problems. This reality is more present when the government is required to take action for or pay attention to impoverished people. Indeed, local government in a small lake is ordinarily the only jurisdiction, explicitly the crux of decision making and executive functions. The essence of the decision making and executive functions is the people, whether living, working or using the small lakes. This is highlighted in a dilemma when the people have diverse and contradicting interests in the lake, so the government will decide in the end. Government is required to a greater extent when there is a deficit or imbalance of entitlement among groups of people in a lake—usually, one group is stronger than another group, who are weaker in staking a claim. In the resolution, the government may be right or wrong due to the presence of sundry factors, such as information, preference, voices and suffrage. If right, it must sustain and be resolute; if wrong, it must adjust or change; however, the government will be the one to determine lake governance. Thus, eventually, the government's function cannot be taken away, as it is inalienable and necessary for a small lake's people.

A lake's people and the government are intensely intertwined in lake governance. The people are the embodiment of human pursuits in the lakes, whether those be undertakings, activities or impediments. Ideally, the government can be excluded, as the collective are the ones to decide for the lake's people. In the real world the government cannot be omitted, since the more people that utilise and develop a lake the more people generate and complicate issues. Thus, as the populace confront predicaments, political authority is

employed more in lakes. In addition, the collective people are typically composed of groups with distinct interests and circumstances in lakes. Therefore, cooperation and collaboration in tackling issues or decisions of a lake's people are tricky, particularly in making solutions or changing rules, as the dominant interest normally controls and prevails (see Tsebelis 1995; Haggard and McCubbins 2001; Brillo 2011a, 2011b, 2012). If the dominant interest in the small lake is correct, then the people and the government can uphold it. If the dominant interest is mistaken or erroneous, then the government is always the most apt to rectify the issue for a lake's people. In fact, the government (against the other actors) is the most capable and enduring in counteracting the manipulation of captured interests in lakes. When the people are affected by a small lake's hindrances, then the government will be the one to settle and counterbalance.

### 3. Government and Premier Leader in Small Lakes' People

The government is foremost among the constituents in lake governance. However, the government's study of lakes is scant, as its concept and function are usually examined less thoroughly (e.g., Google Scholar 2021a). This is particularly true for small lakes, where the government is often recognised but meagrely concentrating and committing assiduously (e.g., Brillo 2015a, 2016e, 2020b). Accordingly, adhering to lake governance, the concept of the government directly signifies the primary institution, where its premier function steers political executive authority and decisive administrative power in a lake's people. Political executive authority solely refers to the definitive and legitimate capacity to put in place and enforce policies and regulations in lakes. The decisive administrative power plainly means the determinative and recognised capability to process or the action of making consequential choices vis-a-vis the groups of a lake's people. In other words, the government is the central authority and the dominant power in a lake's rules of the game. Although there are other governmental responsibilities, executive and decision-making activities are the fundamental responsibilities for small lakes, especially in terms of economic development and ecological conservation. For this reason, the concept of the government is reduced into the two conceptualisations following parsimony's principle, the simplest way to explain and understand them. Put simply, a small lake's people are the main function of the government, operated as political executive authority and decisive administrative power.

Political executive authority and decisive administrative power are equated to the premier leaders in lakes (see Figure 2). The executive and administrative institution is the focus, as the bureaucratic agencies are securely attached to the premiership—the authoritative individual person of a lake's people (see Brillo 2013, 2014b). The premier leader is firmly connected to the local government (e.g., the mayor or governor), since the authority and power are consistently top-down in dealing with issues in small lakes. The government's top-down outlook has considerable empirical evidence, as customarily the premier leader presides over small lakes (see Brillo 2015a, 2015b, 2015c, 2016a, 2016b, 2016c, 2016d, 2016e, 2017a, 2017b, 2017c, 2017d, 2020a, 2020b). The concept of the premier leader is primarily about the critical aspects of resoluteness and decisiveness in small lakes. There are plentiful and diverse leaderships traits in the literature, but among a small lake's people the rudimentary components of resoluteness and decisiveness are typically underscored by the government (see Haggard and McCubbins 2001; McCubbins and Cox 2001; Brillo 2011a). Resoluteness refers to the ability of the premier leader to commit and sustain the rules and policies in lakes. Decisiveness refers to the ability of the premier leader to settle and determine the rules and policies in lakes. In lakes, resoluteness normally means initiating or establishing and maintaining concerning development or conservation, while decisiveness means finding or solving and making decisions concerning the problem. Overall, the premier leader's exemplar is ample in small lakes, where the government dominates and engages in lake governance (e.g., International Lake Environment Committee Foundation 2005; Nakamura and Rast 2014; Brillo 2017b, 2017d).

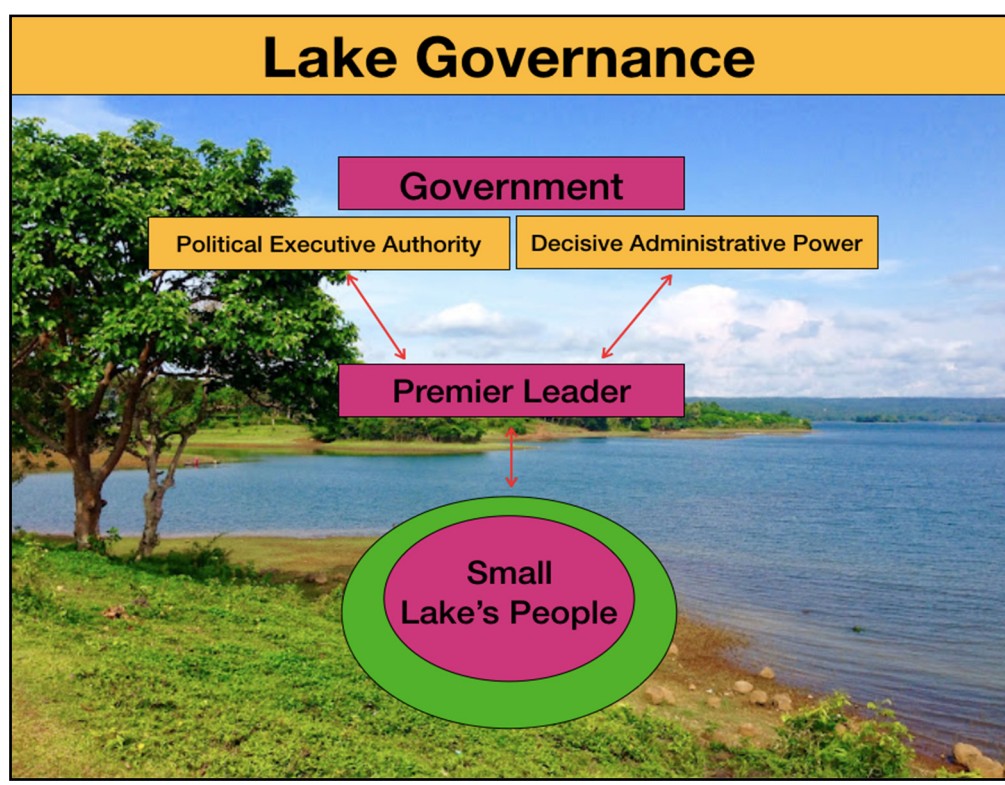

**Figure 2.** Lake governance: government, premier leader and small lake's people.

The premier leader necessitates effective government in lakes. This is crucial since producing the intended outcome is usually measured in success. An effective government's indicators are numerous in the literature, but a few mention consolidating and explicating its concept vis-a-vis small lakes. An effective government is generally associated with addressing economic development and ecological conservation in lakes as well as confronting the contending multiple and diverse interests in their peoples. The notion of an effective government is manifested and encapsulated in the known Integrated Lake Basin Management's (ILBM) six fundamental pillars for governance—institutions, policies, participation, information, technology and finance, which are captured within the concept of a premier leader (see Nakamura and Rast 2014; International Lake Environment Committee Foundation 2021). This is emphasised since if part of the pillars is absent or deficient in a small lake, then the premier leader can assume it and eventually consolidate them. Ideally, the premier leader should be a genuine concern for an effective government in lakes. When the people are unanimous in a small lake, then that is uncomplicated and acceptable—the premier leader will just implement. However, again, when the people are fragmented or there are many with various interests in a small lake, then that is intricate and challenging. That is why, in the real world, the premier leader is critical in asserting to compel the collective people in a small lake.

The premier leader must be present and active in small lakes. This is paramount due to the immense obligation and responsibility being conventionally a public goods as a lake and its people. The government's leader is the equilibrium among the different interests of a lake's people, from individuals and groups to businesses and environments. On the other hand, if the premier leader is inadequate or ineffective, the other actors (i.e., non-governmental or bureaucratic actors) are expected to shoulder accountability in a small lake. Typically, this is carried out when the local government is exceedingly occupied with other things (e.g., financial problems—urban poverty/unemployment or an imbalanced budget), making it hard to concentrate on a lake. However, this arrangement (with the other actors) is provisional, since the premier leader, in time, will seize on the small lake. This can be attained by either the government's leader refocusing, or a new leader (usually via

election) emerging in a small lake. Whether refocusing or emerging, the premier leader's pivotal role is to tackle the obstacles and make the small lake and its people operational with a long-term commitment. Thus, the premier leader eventually cannot be ignored and will exist in the government in the long run.

Conversely, if the premier leader has been functioning and prevailing in small lakes but lost in the election, then there are dual anticipated probabilities. First, the new leader can continue or persevere with the inception of policies and activities of a lake and its people. This prospect is the stability in the intention of the small lake's people. Second, the new leader can change or shift the policies and activities that have started in a small lake. This scene is the enhancement in the expectation of a small lake's people. Whatever the probability, as long as the premier leader attends and remains in a small lake, it creates more incentives to maintain. Therefore, whether continuing or changing, the government's leader is obliged to stay on via increasing returns—locking in from the past leader to the present leader (see Levi 1997; Pierson 2000; Brillo 2008, 2014a). In other words, it is difficult to negate once the premier leader has started in a small lake, as the subsequent government leaders are compelled to follow. This is due to the fact of path dependence, particularly the current status built by the previous leader, which binds them to the succeeding government leaders. Thus, when the premier leader has embarked on the course or track of a small lake, gradually but steadily, it becomes institutionalised, making it reinforcing and accepting.

The premier leader is the key; whatever the lake governance's deficiency, inasmuch as the government leader is existing and engaging in a small lake, then slowly and steadily it shall impute a constructive outcome. If missing, the government is difficult to emerge in a lake, but it often becomes more feasible and manageable when already active. Even in the worst predicament, as long as the premier leader is present in a small lake, it will somehow contribute to the collective people. A small lake should never be apart from the government, as drawbacks (or successes) have multiple causes (or grounds), but it is the most among them. Politics in lakes, especially small lakes, is critical, as the premier leader is the most considerable influence among the general public and the most deliberate effort on a small lake's people. Overall, the government is the overarching principle; explicitly, political executive authority and decisive administrative power dominate lakes and their peoples. The course of lake governance, the government, and the premier leader's configuration should be from insecurity to predictability in a small lake's people. Learning, sharing and reproducing among the countless small lakes and their peoples' living should be strengthened and persistent in the direction of the future. In the end, the premier leader ought to be the irreversible attribute and trajectory of a small lake and its people.

## 4. Concluding Remarks

Lake governance is the foremost and essential conceptualisation, the government is a compulsory and paramount constituent and the premier leader is an imperative and obligatory demand in a small lake and its people. Overall, this article asserts that social sciences in small lakes should recognise, enhance and elevate the concept of lake governance by explicitly emphasising the government's principle and embracing the premier leader's conviction. Firstly, the people, small lakes and social sciences are tied up by discussing and clearing the lakes–populace, the large lakes–small lakes and natural sciences–social sciences disputations. Social-sciences-based works are about the study of enhancing the collective people of lakes, precisely small lakes' communities, whether of residents, settlers and/or townspeople. The scholarly outputs on small lakes' peoples are fewer, and mostly distinctly peripheral in the sphere of social sciences studies (compared to natural sciences studies and large-lake studies), thus restricting the big picture. Consequently, the social sciences ought to be dedicatedly secured to small lakes, intensely bridged to natural sciences and manifestly interconnected to large lakes in engaging, understanding and complementing the advancement from the present into the future.

Secondly, lake governance, the government and a small lake's people are connected by understanding and considering the conceptualisation of lake governance and its profound

association with the government and a lake's community. Lake governance is the central concept in lakes, yet it has lesser cognisance and nominal discourse, particularly in the literature on small lakes. In essence, lake governance relates to undertaking authority and power by operating the administration and management of small lakes and their peoples, especially in economic development and ecological conservation. Wholly and purposely, lake governance is primarily about the government supervising and/or intervening in the collective populace of small lakes. Thirdly, the government and premier leader of a small lake's people are bridged by the course of lake governance by highlighting and asserting the government's overarching principle and the premier leader's pivotal ascendancy of concepts in a small lake's collective community. The government's study of small lakes is frequently conceded but modestly concentrated and perpetuated rigorously. The government is the premier leader's central obligation and responsibility that directs the political executive authority and decisive administrative power of a small lake's people. In summary, the concept of a premier leader is fundamentally about the crucial features of resoluteness, establishing and maintaining the rules of a small lake, and decisiveness, resolving and determining the problems of the people. On the whole, a small lake's people are the function of lake governance, where the government and premier leader are employed capably and effectively.

In the state of nature in lakes, the government is not needed, but when society exists the government's premier leader is necessary. When the people begin and escalate in a small lake, the collective community's goals are often inconsistent (at worst, contradictory), particularly in terms of the utilisation and exploitation of development versus the preservation and restoration of conservation. In staying and living in a small lake, the people are ordinarily biased to development and secondary to conservation at the start. Conservation is usually evolved after the people have some stability of subsistence in a small lake. However, whatever the arrangement in a small lake's people, equipoising the collective community's goals are difficult and sometimes burdensome, as the interests are usually more than one and diverse. With this balancing act, the government (among the other actors) is mandatory and required to pursue the purpose of a small lake; at the end of the day, no one else but the government shall have to function. However, for the administration of a lake, the premier leader is the highest form of performance and service. The premier leader is the unparalleled government's equilibrium in flourishing the development of the people and protecting the conservation of a small lake. It is the predominant outcome and the most efficacious way to address issues, make decisions and improve lakes. Thus, lake governance matters—it is about the government and premier leader, as a small lake's people are always political.

**Funding:** This research received no external funding.

**Informed Consent Statement:** Not applicable.

**Data Availability Statement:** Not applicable.

**Conflicts of Interest:** The author declares no conflict of interest.

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
