# Peer review of "Government, Premier Leader and Small Lakes’ People Vis-a-Vis Lake Governance"

_socsci, doi:10.3390/socsci11040165_

Round 1
Reviewer 1 Report
First, my thanks and congratulations to the authors for submission of their work that I find both fascinating and timely. I can see that the authors address the issues of political autonomy while recognizing the need for linking political autonomy with the wider political milieu that holds the fabric of a governed society together.
I enjoyed the arguments that posited the importance of specific leadership figures as well as the lucid explanations of the rather intricate relationships between the various concepts presented in the paper.
If I may, there are a few small items that need to be cleaned up via a minor revision, and they are:
Page 2 Line 48: Do you mean to say "Water is the most important but causes a predicament for the lake people...."? If so, please correct this. Thank you.
Page 14 Lines 497-499: Do you mean "via a long-term commitment" and "will exist in the Government in the long run"? If so, please correct this. Thank you.
Page 15 Line 533: Is it "Concluding remarks" or "Conclusion remarks"? Please amend if necessary. Thank you.
A note: There is an overwhelming dependence on the work of Brillo in the citation and reference. There's nothing specifically wrong with that as the works of other authors are also cited, but perhaps in future the authors may want to not have so heavy a dependence on one author. Thank you.
Just a small note on APA, when having more then one source in parenthesis please separate the sources by using a semi-colon and not a comma. Thank you.
Overall, a fascinating read and best of luck to the authors.
Author Response
Reviewer 1
I am sending the revisions of my manuscript, entitled “Government, Premier Leader and Small Lake’s People vis-a-vis Lake Governance.”
The following are the responses (in red color):
- Page 2 Line 48: Do you mean to say "Water is the most important but causes a predicament for the lake people...."? - “Water use is the most important but causes a predicament for the lake’s people”
- Page 14 Lines 497-499: Do you mean "via a long-term commitment" and "will exist in the Government in the long run"? - “Whether refocusing or emerging, the Premier Leader’s pivotal role is to tackle the obstacles and make the small lake and its people operational via a long-term commitment. Thus, the Premier Leader eventually cannot be ignored and will exist in the Government in the long run.”
- Page 15 Line 533: - “Concluding Remarks”
A note: There is an overwhelming dependence on the work of Brillo in the citation and reference. There's nothing specifically wrong with that as the works of other authors are also cited, but perhaps in future the authors may want to not have so heavy a dependence on one author. - Yes, I use Brillo’s references extensively since he is the “authority” in small lake’s social science study; besides, Brillo (as I know) is the sole specialist (i.e., devoid of authors or writers) in lake governance on a small lake in the Philippines.
Just a small note on APA, when having more then one source in parenthesis please separate the sources by using a semi-colon and not a comma. Thank you. - Already done
Best,
Author(s)
Reviewer 2 Report
The conceptual meaning of the document is relevant and the authors present the work in an interesting way. However, the scientific significance and academic relevance are not clear. Additionally, the objective vanishes after reading the article, it becomes a not so clear search to prove the scant importance that the scientific community gives to the study and relevance of small lakes.
Authors write several sections of text in italics, is it necessary?
The figures are not necessary and the way they are presented makes no sense.
Author Response
Reviewer 2
I am sending the revisions of my manuscript, entitled “Government, Premier Leader and Small Lake’s People vis-a-vis Lake Governance.”
The following are the responses:
- The conceptual meaning of the document is relevant and the authors present the work in an interesting way. - This is strongly pertinent since the paper is about the meaning of the concepts of Lake Government and Government (particularly Premier Leader) vis-a-vis small lakes. I have argued that the word— Lake Governance’s concept is deficient and esoteric at present. The terms have a minimal discussion on the central concept in lake literature, making it insufficient attention in the discourse. I am an intellectual type (i.e., Political Philosophy or Theory), so if you want to understand how I write, please read the following works: W. Ebenstein and A. Ebenstein (Great Political Thinkers), Plato (Republic), or Jean Jacques Rousseau (The Social Contract).
- The scientific significance and academic relevance are not clear. - That is clear since this paper is about a literature review by connecting Small Lakes (versus Large Lakes), Social Sciences (versus Natural Sciences), Lake Governance (versus Governance), and Government, then showing the literature gaps, and lastly discussing and explaining the heart of the Government— the crucial Premier Leader in small lakes.
- Authors write several sections of text in italics, is it necessary? - Yes, I want to emphasize the central premises or crucial statements.
- The figures are not necessary and the way they are presented makes no sense. - Yes, the two figures are essential; this journal (i.e., Social Science Basel) is a hybrid publication, which will help the other academic disciplines understand my work.
Best,
Author(s)
Round 2
Reviewer 2 Report
I do not see any substantial improvement in this revised version. In fact, I can see mostly format changes.
I appreciate the author's conviction about his/her work. However, the way he/she presented it is not clear. Italics and graphics are unnecessary and make the article not easy to read and understand.
Author Response
Dear Reviewer 2,
I am sending my reply on Round 2. The following are the responses (in red color):
- The way he/she presented it is not clear. - It is comprehensible; please read the end part of the introduction: "Overall, this writing discusses and urges to address the critique on small lakes and Lake Governance. The article aspires to look into the central concept for the small lake's people in social sciences, particularly Lake Governance, its most crucial constituent— Government, and its most decisive function, Premier Leader. The main argument is that social sciences in lakes, especially small lakes, must accept, broaden and elevate Lake Governance by focusing and embracing the concept of Government and Premier Leader. The study encompasses the literature gaps to connect, enhance and expand the ideas of social sciences in Lake Governance vis-a-vis Government, Premier Leader and the people to have a bigger picture in the small lakes. The conceptualisation among them are mostly wanting, less attached and seldom scrutinised in the academic literature. The compositional concepts are confronted and synthesised to amplify the knowledge base in social sciences and situate the Lake Governance in lake scholarship. This is part of the deep-rooted desire to explore and perpetuate the academic rationale in small lakes. Therefore, this paper delineates the discourse as follow: above, (a) Lake's People, Small Lakes and Social Sciences; below, (b) Lake Governance to the Government and Small Lake's People; (c) Government and Premier Leader in Small Lake's People; and (d) Concluding Remarks.”
- Italics and graphics are unnecessary and make the article not easy to read and understand. - On the italics, although that will help the audience by emphasizing the central premises or crucial statements, but if the Editor-in-Chief would like to remove that (Italics), then I will agree with that (the reviewer 2 comment is just a suggestion, so I follow the instruction of the Editor-in-Chief and the comment of the reviewer 1). On the graphics (i.e., Figures 1 and Figure 2), I strongly argue that they are essential in my article. The Social Science-Basel is a journal that encompasses a wide array of academic disciplines, so the two Figures will help the other disciplines understand my work. Plus, the reviewer 2 comment is just a suggestion, so I will wait for the direction of the Editor-in-Chief and the comment of the reviewer 1.
- The conceptual meaning of the document is relevant and the authors present the work in an interesting way. - This is strongly pertinent since the paper is about the meaning of the concepts of Lake Government and Government (particularly Premier Leader) vis-a-vis small lakes. I have argued that the word— Lake Governance's concept is deficient and esoteric at present. The terms have a minimal discussion on the central concept in lake literature, making it insufficient attention in the discourse. I am an intellectual type (i.e., Political Philosophy or Theory), so if you want to understand how I write, please read the following works: W. Ebenstein and A. Ebenstein (Great Political Thinkers), Plato (Republic), or Jean Jacques Rousseau (The Social Contract).
Best,
Author(s)
Round 3
Reviewer 2 Report
In order to review an improved version of this work, authors must omit some parts that are unnecessary: italics and graphics. I strongly ask for it, in order to help readers to understand better this paper as they make it unclear.